# Time trends in perinatal outcomes among HIV-positive pregnant women in Northern Tanzania: A registry-based study

Tormod Rebnord[1]*, Blandina Theophil Mmbaga[2,3], Ingvild Fossgard Sandøy[1,4], Rolv Terje Lie[1,5], Bariki Mchome[3,6], Michael Johnson Mahande[3,7], Anne Kjersti Daltveit[1,8]

1 Department of Global Public Health and Primary Care (IGS), Faculty of Medicine, University of Bergen, Bergen, Norway, 2 Department of Paediatrics and Child Health, Kilimanjaro Christian Medical University College, Moshi, Tanzania, 3 Kilimanjaro Clinical Research Institute, Kilimanjaro Christian Medical Centre, Moshi, Tanzania, 4 Centre for Intervention Science in Maternal and Child Health (CISMAC) and Centre for International Health, University of Bergen, Bergen, Norway, 5 Centre for Fertility and Health, Norwegian Institute of Public Health, Oslo, Norway, 6 Department of Gynecology and Obstetrics, Kilimanjaro Christian Medical University College, Moshi, Tanzania, 7 Department of Epidemiology & Biostatistics, Institute of Public Health, Kilimanjaro Christian Medical University College, Moshi, Tanzania, 8 Department of Health Registry Research and Development, Norwegian Institute of Public Health, Bergen, Norway

* tormod03@gmail.com

**Data Availability Statement:** All relevant data are within the paper and its Supporting information files.

## Abstract

### Introduction

Maternal HIV infection is associated with increased risk of having a preterm delivery, low birth weight baby, small for gestational age baby and stillbirth. Maternal use of combination antiretroviral treatment is also associated with preterm delivery and low birth weight, although the effects vary by the type of drugs and timing of initiation.

### Objective

To examine time trends in adverse perinatal outcomes among HIV-positive compared with HIV-negative women.

### Design

Registry-based cohort study.

### Setting

Northern Tanzania, 2000–2018.

### Study sample

Mother-baby pairs of singleton deliveries (n = 41 156).

### Methods

Perinatal outcomes of HIV-positive women were compared with HIV-negative women during time periods representing shifts in prevention of mother-to-child transmission guidelines.

**Funding:** The authors received no specific funding for this work.

**Competing interests:** The authors have declared that no competing interests exist.

Monotherapy was used as first-line therapy before 2007 while combination antiretroviral treatment was routinely used from 2007. Log binomial and quantile regression were used to analyze the data.

## Main outcome measures

Preterm delivery, low birth weight, perinatal death, stillbirth, low Apgar score, transfer to neonatal care unit and small for gestational age.

## Results

Overall, maternal HIV infection was associated with a higher risk of low birth weight and small for gestational age. Moreover, this pattern became more pronounced over time for low birth weight, the last time period being an exception. For other outcomes we found none or only a small overall association with maternal HIV infection, although a trend towards higher risk over time in HIV-positive compared with HIV-negative women was observed for preterm delivery and perinatal death. Quantile regression showed an increase in birth weight in babies born to HIV-negative women over time and a corresponding decline in birth weight in babies born to HIV-positive women.

## Conclusion

Unfavourable trends in some of the selected perinatal outcomes were seen for HIV-positive compared with HIV-negative women. Potential side-effects of combination antiretroviral treatment in pregnancy should be further explored.

## Introduction

The HIV pandemic affects individuals in every continent of the world. Globally, 37.7 million people were living with HIV in 2020, and 53% among them were women and girls [1]. Despite a 55% decline in new HIV infections in Eastern and Southern Africa from 2000 to 2020, these regions are still severely affected with 670 000 new cases reported in 2020 [2], which constituted 45% of new HIV infections globally this year. Maternal use of combination antiretroviral treatment (cART) has the potential to virtually eliminate mother-to-child transmission (MTCT) of HIV if started before pregnancy [3], as compared to a 15% to 45% risk of MTCT without treatment [4]. The percentage of pregnant women living with HIV who received antiretrovirals for prevention of MTCT (PMTCT) in Eastern and Southern Africa has increased gradually from 52% in 2010 to 91% in 2020 [2]. Correspondingly, the annual number of children (0–14) newly infected with HIV declined from 210 000 in 2010 to 75 000 in 2020.

Maternal HIV infection has been associated with an increased risk of having a preterm delivery, a low birth weight (LBW) baby, a small for gestational age (SGA) baby, and stillbirth [5, 6]. Studies have also indicated a possible higher risk of neonatal death and perinatal death [5, 7]. Furthermore, in a Chinese study, maternal HIV infection (presumably mostly untreated women) was associated with an increased risk of a low Apgar score at five and 10 minutes after birth [8], and in another study from Nigeria, untreated HIV-positive women were found to be at higher risk of low 5-minutes Apgar score compared with women on cART [9]. Studies have also showed a higher risk of the baby being admitted into the neonatal unit if the mother was

infected but untreated compared with on cART [9], and if the mother was HIV-positive compared with HIV-negative [10].

Maternal use of cART has also been reported to be associated with an increased risk of having a preterm delivery or a LBW baby when compared to monotherapy, although the evidence is conflicting and the effects vary by the type of drugs and timing of initiation [11–18]. An increased risk of stillbirth and having a SGA baby was found among women continuing cART initiated *before* pregnancy when compared with all other HIV-infected women (including those who initiated monotherapy, cART or no ART *during* pregnancy) [18]. A study from Kilimanjaro Christian Medical Centre (KCMC) in Northern Tanzania, using data between 1999 and 2006 from the same birth registry as in this study, found an increased risk of having a baby with low Apgar score when treated HIV-positive women were compared with HIV-negative women [6], although another study from Nigeria found that the risk of perinatal mortality, low 5-minutes Apgar score and admission into neonatal unit was comparable between HIV-positive women on cART and HIV-negative controls [19]. Nevertheless, preterm delivery and LBW are much more frequently studied outcomes than the other perinatal outcomes mentioned.

Recommendations on treatment options and eligibility criteria for pregnant women, based on levels of CD4 cell count and clinical stage, have been updated several times by the World Health Organization (WHO). Many resource-limited countries have based their PMTCT guidelines on the WHO recommendations [20–24]. First-line therapy has since it was introduced by WHO in 2004 consisted of two nucleoside/nucleotide reverse transcriptase inhibitors (N(t)RTIs) combined with a non-NRTI (NNRTI) as a third agent. However, monotherapy was still used as first-line therapy for many years during the 2000s in low- and middle-income countries, before triple-drug therapy, known as cART, became available. Most health facilities in Tanzania, including KCMC, did not implement cART routinely for pregnant women before 2007 [25]. Pregnant women living with HIV who received cART for PMTCT in Tanzania increased from 48% in 2010 to 84% in 2020 [2].

In this study, we examined time trends in adverse perinatal outcomes among HIV-positive women compared with HIV-negative women in Northern Tanzania during an 18-year time period, spanning shifts in PMTCT guidelines and antiretroviral treatment coverage among the included women.

## Methods

### Design and setting

This was a registry-based cohort study which was designed using maternally linked data from Kilimanjaro Christian Medical Centre (KCMC) birth registry. KCMC is a tertiary referral hospital located in Moshi Urban District, Northern Tanzania. Moshi is the capital of the Kilimanjaro region, a region of seven districts and a total population of more than 1.6 million people as of 2012 [26]. Only 24.2% of the Kilimanjaro population lives in urban areas. Tanzania was categorized as a lower middle-income country in 2020, with a life expectancy at birth of 65 years in 2018, an increase from 50.8 years in 2000 [27].

KCMC is a non-profit private hospital, and as one of four zonal hospitals in Tanzania, KCMC is a referral hospital for more than 15 million people in the northern part of Tanzania [28]. Therefore, the Gynecological and Obstetric Department receives high risk cases from the nearby regions. While inpatient services such as admittance and surgical procedures incur cost-sharing fees, antiretroviral treatment (ART) for prevention of mother-to-child transmission (PMTCT) has been free of charge since September 2004. Also, this was the year when KCMC implemented national PMTCT guidelines based on WHO guidelines. Antenatal

services at KCMC have been free of charge during the whole study period, except for some services and investigations beyond general antenatal care and PMTCT care, where cost-sharing fees or health insurance support were needed.

## Data collection and recruitment

The KCMC medical birth registry has been in operation since July 2000 [29]. Around 2500–4500 deliveries are registered annually, and about 20% of these are referred from other health facilities. Each record in the registry represents one birth, i.e. one child, live born or stillborn. For each delivery a unique delivery identification is assigned, i.e. twins and higher order multiple births are assigned the same delivery identification number. The mother is also assigned a hospital number, which remains the same if the mother is having a successive delivery. The data are collected from medical records where all women giving birth at KCMC are interviewed within 12 to 24 hours after each delivery. The interview is conducted in swahili by trained nurse-midwives, using a structured questionnaire. Mothers receive information about the registry and its contents, and information is only collected from mothers who have provided oral consent. Nearly all women who deliver at KCMC are included in the registry. Maternal HIV status (positive or negative) is registered based on the result of the point of care HIV test performed during antenatal care. Although pregnant women are usually tested during the first antenatal care visit, we could not determine how many this applied to as the registry does not include any information about the timing of the HIV test. Further details about the birth registry and recruitment are described elsewhere [29, 30].

## Data selection

We only included singletons born between 2000 and 2018 with birth weight between 500 and 5000 grams. Twins and higher order multiple births were removed as these infants would have a higher risk of several of the adverse perinatal outcomes compared to singletons. In addition, we only included deliveries of women living in the local catchment area of Moshi (Moshi urban and rural).

## Study design and variable definitions

**Main exposure.**   Positive maternal HIV status was the main exposure in this study. We divided the variable maternal HIV status into three categories: Positive, negative and unknown HIV status.

**Time periods.**   We defined a time period variable, based on time points when changes in national PMTCT guidelines in Tanzania occurred, for stratification of the analysis [25, 31–36]:

The time period 2000–2003 represents the pilot phase before the implementation of national PMTCT guidelines [36]. Pilot nevirapine (NVP), a non-nucleoside/nucleotide reverse transcriptase inhibitor (NNRTI), was the drug of choice [37].

The second time period 2004–2006 corresponds to the introduction and implementation of the first national PMTCT guidelines, adapting the WHO 2004 guidelines [25]. In this second time period, single dose NVP and short course zidovudine/azidothymidine (ZDV/AZT) were the two treatment regimens used for PMTCT purposes in most health facilities in Tanzania. The NVP prophylaxis regimen was used at KCMC, which consisted of a single tablet taken at the onset of labour for all HIV-positive women. Also, provider-initiated testing of HIV status was implemented from 2004, meaning that the health care provider recommended HIV testing for pregnant women during antenatal care or at delivery.

In 2007, the national PMTCT guidelines were revised for the first time, defining the start of the third time period from 2007 to 2011 [31]. These were based on the revised WHO guidelines

from 2006. Triple-drug antiretroviral treatment, known as combination antiretroviral treatment (cART), was introduced as the standard treatment regimen for all women with either an HIV infection classified as WHO stage 4, WHO stage 3 and CD4 <350, or CD4 <200 regardless of WHO stage. This regimen included ZDV/AZT, lamivudine (3TC) and NVP (thus two nucleoside/nucleotide reverse transcriptase inhibitors (NRTIs) and one NNRTI).

In the fourth time period, from 2012 to 2014, the WHO option A was implemented in the national PMTCT guidelines in Tanzania [32]. All women with an HIV infection classified as WHO stage 3 or 4, or CD4 count ≤350, should receive cART. The first-line treatment for pregnant women was ZDV/AZT, 3TC and NVP or efavirenz (EFV).

National PMTCT guidelines implementing WHO Option B+ were introduced in 2014, defining the start of the fifth time period [33, 35]. All HIV-positive women, regardless of clinical stage or CD4 count, are offered lifelong cART, composed of tenofovir (TDF), 3TC and EFV. This should be started immediately after diagnosis, irrespective of pregnancy status.

**Outcome variables.** Indicators of adverse perinatal outcomes were extracted from the birth registry. These included preterm delivery, low birth weight (LBW), perinatal death, stillbirth, low Apgar score, transfer to neonatal care unit and small for gestational age (SGA). Additionally, birth weight as a continuous variable was included as an outcome measure, rounded to the nearest 50 grams when data were entered into the birth registry. Preterm delivery was defined as delivery before completed 37 weeks of gestation. LBW was defined as less than 2500 grams. Perinatal death included both stillbirths and early neonatal deaths, where early neonatal death was defined as death within the first seven days of life. Two variables that were used for defining early neonatal death were missing in the last time period, and estimates for perinatal death were therefore not calculated in the last time period. Stillbirths included both deaths before and during labor. Low Apgar score was defined as an Apgar score from 1 to 6, recorded at any of the three times of assessment (1, 5 or 10 minutes after birth). Transfer to the neonatal care unit was a dichotomous variable that defined whether the newborn was transferred to the neonatal care unit at the pediatric department or not. SGA was defined as birth weight below the sex-specific 10$^{th}$ percentile according to gestational age, using international standards [38]. SGA was defined for gestational week 33 to 42.

**Other variables.** Variables for maternal characteristics were divided into categories. Semi-urban residence was included in the urban category. As we did not have information on the type or number of drugs used as ART during pregnancy, we included a dichotomous variable indicating whether the women were on ART during pregnancy or not.

## Statistical analysis

Categorical variables were summarized using frequency tables and continuous variables using means with corresponding standard deviations, in order to present characteristics of the study sample by categories of maternal HIV status and within each time period. Proportions for parity were not calculated in the last time period as the parity variable was missing in most of the records in the last time period due to changes in the collection process. Numbers and percentages of all outcome variables, including missing observations, are also presented. A stacked bar chart was computed in order to present the distribution of categories of maternal HIV status among all women within each time period. Furthermore, proportions of all outcome variables within categories of maternal HIV status (HIV-positive and HIV-negative) were computed, both overall and within different time periods. We also computed p-values for trends in proportions. Multivariable log binomial regression was used to estimate adjusted relative risks (ARRs) with corresponding 95% confidence intervals for HIV-positive women compared with HIV-negative women. We also calculated p-values for trends in ARRs. Due to the high

proportion of unknown maternal HIV status during the first time period, we excluded this time period from all analyses except for descriptive statistics for sociodemographic characteristics and outcome variables.

A z-score of the continuous variable birth weight at each gestational week from completed 37 weeks to completed 42 weeks was computed, measuring relative size at birth in term deliveries. The z-score was further transformed back to a standardized birth weight. We then performed quantile regression analyses for the standardized birth weight variable in order to assess annual change in birth weight at different percentiles. Results are presented for babies born to all women, HIV-positive women and HIV-negative women. All regression analyses were adjusted for maternal age, marital status, current residence and education level. As the parity variable was missing in most of the records in the last time period, we did not adjust for parity in any of the main regression analyses.

As supplementary analyses, we used the hospital number to calculate robust variance estimates to account for dependencies between maternal siblings. This was not possible for the quantile regression analyses. Furthermore, regression analyses for the second, third and fourth time period were performed with and without adjusting for parity, in order to evaluate the effect of parity not being included as an adjustment variable. We also conducted supplementary analyses of SGA where we excluded preterm deliveries, and we repeated all regression analyses where we excluded HIV-positive women not on ART. Finally, regression analyses for all perinatal outcomes were in addition performed without adjustment for confounders. All data were analyzed using Stata/SE version 17.0.

### Ethical approval

The study was approved by the KCMC College Research Ethics Review Committee, the National Research Ethics Committee in Tanzania (NIMR/HQ/R.8c/Vol. I/1979 dated 16 Feb 2022), and the National Committee for Medical and Health Research Ethics in Norway (reference 2018/354 dated 15 Feb 2019).

## Results

### Characteristics of the study sample

Information about a total of 62 927 babies delivered between 1999 and 2018 were extracted from the KCMC electronic birth registry (Fig 1). A total of 59 337 babies remained after excluding twins and higher order multiple babies. Furthermore, 144 babies delivered in 1999 were excluded, and 198 babies with either missing birth weight or weight outside 500–5000 grams were excluded. Finally, exclusion of women living outside Moshi led to another 17 839 babies being excluded. Thus, the final analysis sample consisted of 41 156 mother-baby pairs of singleton deliveries. HIV-positive women were more likely than HIV-negative women to be a single mother and to have high parity ($\geq$4), rural residence or low educational attainment, both overall and in all time periods (Table 1 and S1 Table). Women with unknown HIV status were more likely than women with known status to have high parity ($\geq$4), rural residence and low educational attainment, both overall and in nearly all time periods (Table 1 and S1 Table).

### Trends by maternal HIV status and maternal use of antiretroviral treatment (ART)

The proportion of women with unknown HIV status among the whole study sample declined steadily, being 73.8%, 30.5%, 6.1%, 1.2% and 1.7%, respectively, in the five time periods (Fig 2). During the same time periods, the proportion of women with positive HIV status among all

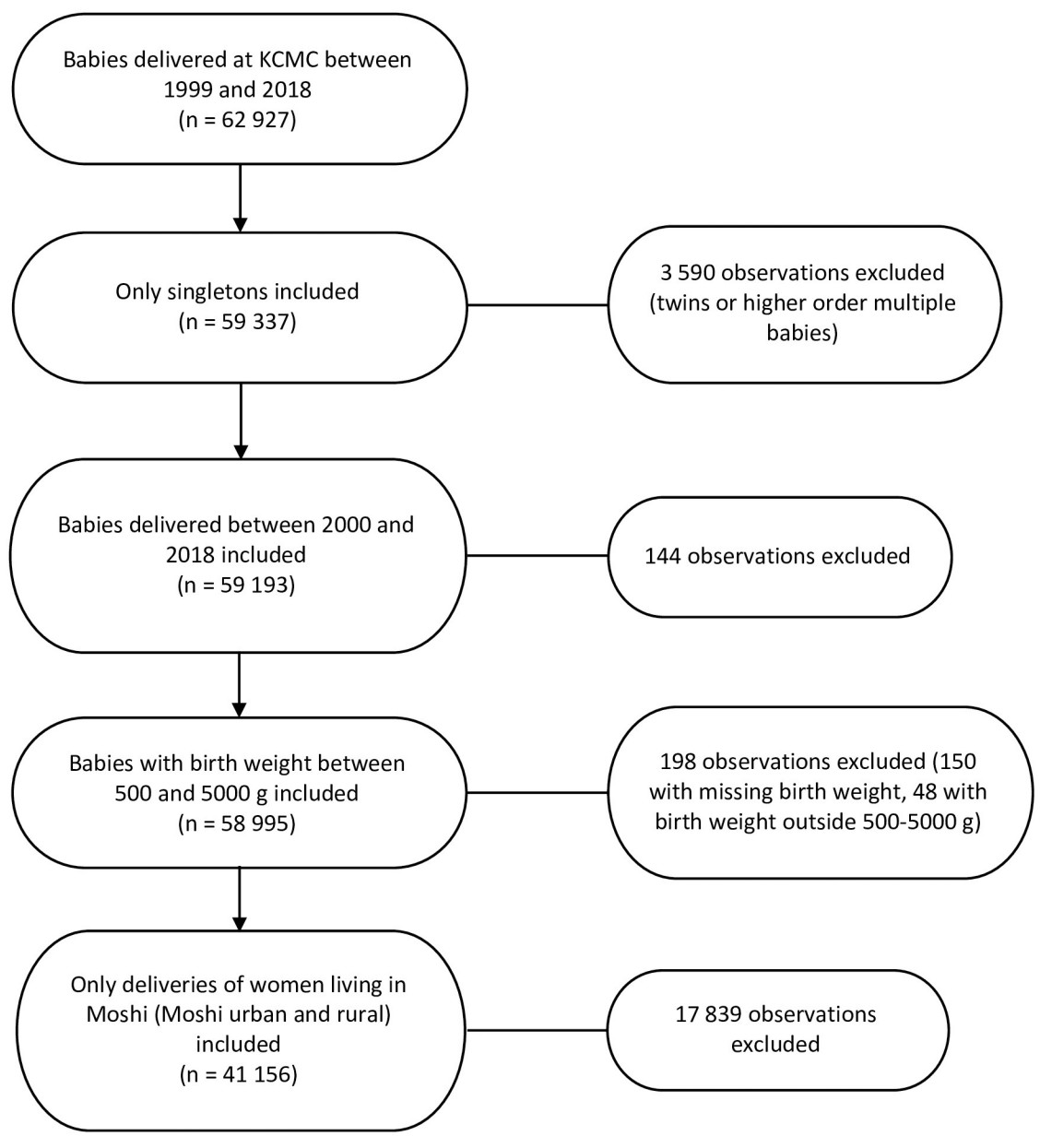

**Fig 1. Final sample flow chart.**

women were 1.9%, 4.8%, 4.6%, 4.6% and 3.8%, respectively (Fig 2). HIV-negative women constituted the remaining 24.4%, 64.7%, 89.3%, 94.2% and 94.5% in the five time periods, respectively (Fig 2). Among the HIV-positive women, the proportion of women on any ART during the five time periods were 60.2%, 79.9%, 89.7%, 88.3% and 84.4%, respectively, with an overall ART coverage of 84.8%.

## Trends in perinatal outcomes

Both for HIV-negative and HIV-positive women the proportion experiencing a preterm delivery increased from the second to the last time period (Table 2). For HIV-negative women

**Table 1.** Left side: Sociodemographic characteristics of study sample, 2000–2018. Right side: Description and definition of outcome variables, 2000–2018.

| Denominators | | N (%) | |
|---|---|---|---|
| Total | | 41 156 | |
| HIV- | | 32 012 (77.8) | |
| HIV+ | | 1683 (4.1) | |
| Unknown HIV status | | 7461 (18.1) | |
| **Sociodemographic** | | | |
| variables | Mean (SD) | Outcome variables | n (%) |
| **Mother's age**[1] | | **Preterm delivery** | |
| HIV- | 28.1 (5.9) | Preterm delivery (gestational week <37) | 4685 (11.4) |
| HIV+ | 29.6 (5.9) | No preterm delivery (gestational week 37–44) | 32 878 (79.9) |
| Unknown HIV status | 26.9 (6.1) | Gestational week < completed 22 or > completed 44 weeks | 2500 (6.1) |
| | | Missing (data not registered) | 1093 (2.7) |
| **Body weight before pregnancy**[2] | | **Low birth weight (LBW)** | |
| HIV- | 64.3 (13.5) | Low birth weight (500–2499 g) | 4060 (9.9) |
| HIV+ | 64.4 (13.4) | No low birth weight (2500–5000 g) | 37 096 (90.1) |
| Unknown HIV status | 61.9 (12.6) | | |
| | % | | |
| **Single motherhood**[1] | | **Perinatal death** | |
| HIV- | 15.0 | Perinatal death | 1472 (3.6) |
| HIV+ | 21.1 | No perinatal death | 39 684 (96.4) |
| Unknown HIV status | 12.9 | | |
| **Nulliparous**[3] | | **Stillbirth** | |
| HIV- | 42.3 | Stillbirth | 1189 (2.9) |
| HIV+ | 30.8 | No stillbirth | 39 967 (97.1) |
| Unknown HIV status | 38.3 | | |
| **High parity (≥4)**[3] | | **Low Apgar score**[4] | |
| HIV- | 12.5 | Low Apgar score | 2121 (5.2) |
| HIV+ | 17.0 | No low Apgar score | 37 659 (91.5) |
| Unknown HIV status | 18.1 | Missing (not registered at any of the three times of assessment) | 1376 (3.3) |
| **Urban residence**[1] | | **Transfer to neonatal care unit**[4] | |
| HIV- | 74.2 | Transfer to neonatal care unit | 5653 (13.7) |
| HIV+ | 72.1 | No transfer to neonatal care unit | 35 384 (86.0) |
| Unknown HIV status | 60.2 | Missing (not registered) | 119 (0.3) |
| **Education ≤ primary**[1] | | **Small for gestational age (SGA)** | |
| HIV- | 47.3 | SGA (defined for gestational week 33–42) | 5010 (12.2) |
| HIV+ | 56.7 | No SGA | 30 411 (73.9) |
| Unknown HIV status | 70.0 | Gestational age outside 33–42 weeks | 4522 (11.0) |
| | | Missing gestational age or sex (not registered) | 1213 (3.0) |

[1] <0.5% missing observations.

[2] 7746 (18.8%) observations missing.

[3] Data on parity only available from 2000 to July 2014. Percentages are calculated within this time period. 0.01% missing observations for parity between 2000–2013.

[4] Only live births in denominator.

there were also clear trends in the proportions with the following outcomes: LBW (increase), transfer to the neonatal care unit (increase) and SGA (decline).

Overall and in nearly all time periods, women with unknown HIV status experienced the highest proportions of preterm delivery, perinatal death, stillbirth, low Apgar score and transfer to neonatal care unit (S2 Table).

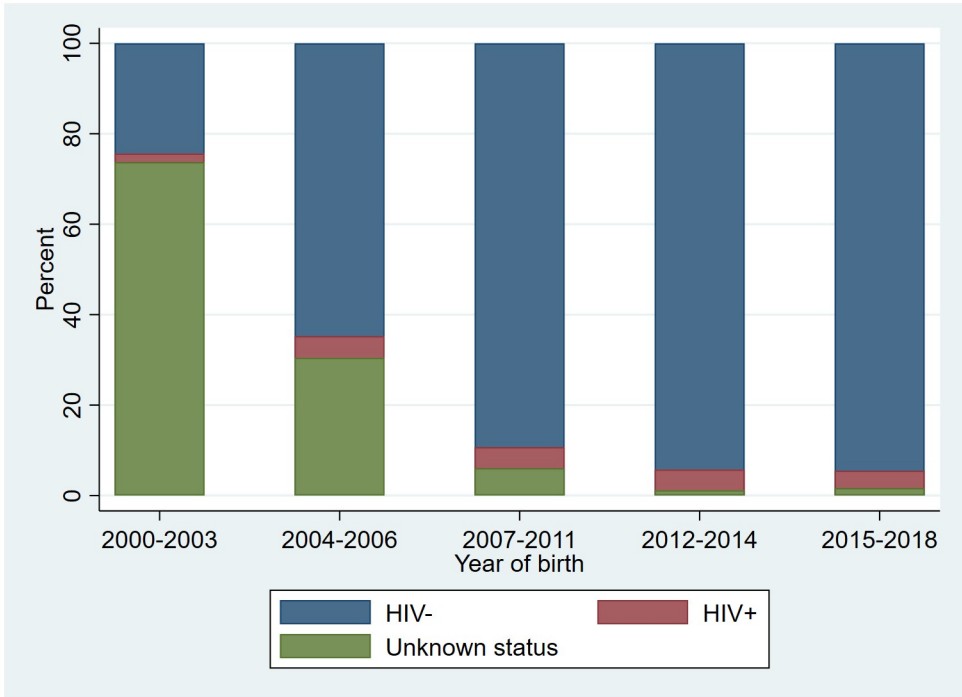

**Fig 2. Distribution of categories of maternal HIV status for the whole study sample within each time period.**

Overall, the highest adjusted relative risks (ARRs) for HIV-positive women compared with HIV-negative women were observed for LBW (1.33, 95% CI 1.18–1.49) and SGA (1.38, 95% CI 1.24–1.53) (Fig 3). An increase in the ARR from the second time period (2004–2006) to the fourth time period (2012–2014) was observed for preterm delivery (p = 0.05), LBW (p = 0.28), perinatal death (p = <0.01) and stillbirth (p = 0.55). Furthermore, the ARRs declined from the fourth time period (2012–2014) to the last time period (2015–2018) for these outcomes except for perinatal death for which data were missing in the last time period.

## Quantile regression

We observed an annual increase in birth weight at all percentiles for babies born to HIV-negative women, with larger increases at the higher percentiles (Table 3). On the other hand, we noted an annual decline in birth weight at all percentiles except the 90th percentile for babies born to HIV-positive women, and the declines were larger at the 10th, 50th and 75th percentile than at the 25th percentile.

## Supplementary analyses

In the robust variance estimates to account for dependencies between maternal siblings, the confidence intervals and p-values for trends in ARRs changed minimally (S3 Table). Overall ARR of LBW (2004–2014) was 1.46 (95% CI 1.28, 1.66) when additionally adjusting for parity, as compared to 1.33 (95% CI 1.18, 1.49) when parity was not included (S3 Table). Apart from this, adjustment for parity did not change the results of the regression analysis substantially for time periods between 2004 and 2014, neither did the results from regression analyses for SGA where preterm deliveries were excluded (S3 Table). For preterm delivery, LBW, perinatal

**Table 2. Adverse perinatal outcomes among HIV-negative and HIV-positive pregnant women in Northern Tanzania, overall proportions and by time period[1], 2004–2018.**

| | 2004–2006 | 2007–2011 | 2012–2014 | 2015–2018 | p for trend | Overall (2004–2018) |
|---|---|---|---|---|---|---|
| Denominators | | | *n (%)* | | | *n (%)* |
| Total | 4113 | 12 565 | 8131 | 7240 | | 32 049 |
| HIV- | 3829 (93.1) | 11 943 (95.0) | 7754 (95.4) | 6958 (96.1) | <0.001 | 30 484 (95.1) |
| HIV+ | 284 (6.9) | 622 (5.0) | 377 (4.6) | 282 (3.9) | <0.001 | 1565 (4.9) |
| **Preterm delivery** | | | | | | |
| HIV- | 358 (10.0) | 1238 (11.6) | 966 (13.3) | 908 (14.3) | <0.001 | 3470 (12.4) |
| HIV+ | 25 (9.5) | 71 (13.1) | 66 (18.6) | 51 (20.7) | <0.001 | 213 (15.1) |
| **Low birth weight (LBW)** | | | | | | |
| HIV- | 347 (9.1) | 1067 (8.9) | 759 (9.8) | 703 (10.1) | 0.01 | 2876 (9.4) |
| HIV+ | 30 (10.6) | 86 (13.8) | 68 (18.0) | 28 (9.9) | 0.71 | 212 (13.6) |
| **Perinatal death** | | | | | | |
| HIV- | 129 (3.4) | 460 (3.8) | 224 (2.9) | – [2] | 0.03 | 813 (3.5) |
| HIV+ | 5 (1.8) | 29 (4.7) | 19 (5.0) | – [2] | 0.71 | 53 (4.1) |
| **Stillbirth** | | | | | | |
| HIV- | 104 (2.7) | 314 (2.6) | 208 (2.7) | 183 (2.6) | 0.90 | 809 (2.6) |
| HIV+ | 5 (1.8) | 21 (3.4) | 18 (4.8) | 7 (2.5) | 0.42 | 51 (3.3) |
| **Low Apgar score** | | | | | | |
| HIV- | 205 (5.5) | 570 (4.9) | 415 (5.5) | 309 (4.6) | 0.17 | 1499 (5.1) |
| HIV+ | 19 (6.8) | 33 (5.5) | 17 (4.7) | 12 (4.4) | 0.18 | 81 (5.4) |
| **Transfer to neonatal care unit** | | | | | | |
| HIV- | 490 (12.8) | 1319 (11.0) | 1146 (14.8) | 1231 (17.8) | <0.001 | 4186 (13.8) |
| HIV+ | 37 (13.0) | 78 (12.6) | 57 (15.2) | 44 (15.7) | 0.19 | 216 (13.8) |
| **Small for gestational age (SGA)** | | | | | | |
| HIV- | 526 (15.5) | 1455 (14.4) | 839 (12.3) | 668 (11.2) | <0.001 | 3488 (13.2) |
| HIV+ | 55 (22.0) | 96 (18.6) | 57 (17.3) | 37 (17.4) | 0.10 | 245 (18.6) |

[1] Time periods: 2004–2006: WHO 2004 guidelines; 2007–2011: Revised WHO 2004 guidelines; 2012–2014: WHO Option A guidelines; 2015–2018: WHO Option B + guidelines

[2] Two variables that were used for defining early neonatal death were missing in the last time period

death and stillbirth, trends in ARRs appeared more evident from the second to the fourth time period in analyses where we excluded HIV-positive women not on ART, and in the quantile regression the annual decline was more evident for HIV-positive women when untreated women were excluded (S3 Table). In regression analyses for all perinatal outcomes, adjustment for confounders did not change the results to a considerable extent (S3 Table).

## Discussion

### Trends in categories of maternal HIV status

During four time periods, defined according to shifts in prevention of mother-to-child transmission guidelines, we found a large decline in the proportion of women with unknown HIV status, an increase in the proportion who had been tested and found HIV-negative, and a stable proportion of HIV-positive women when estimated among all women in our study sample. Unknown maternal HIV status was associated with several indicators of low socioeconomic status. Women with unknown HIV status had the highest risk of all adverse perinatal outcomes, except for low birth weight (LBW) and small for gestational age (SGA), but for low

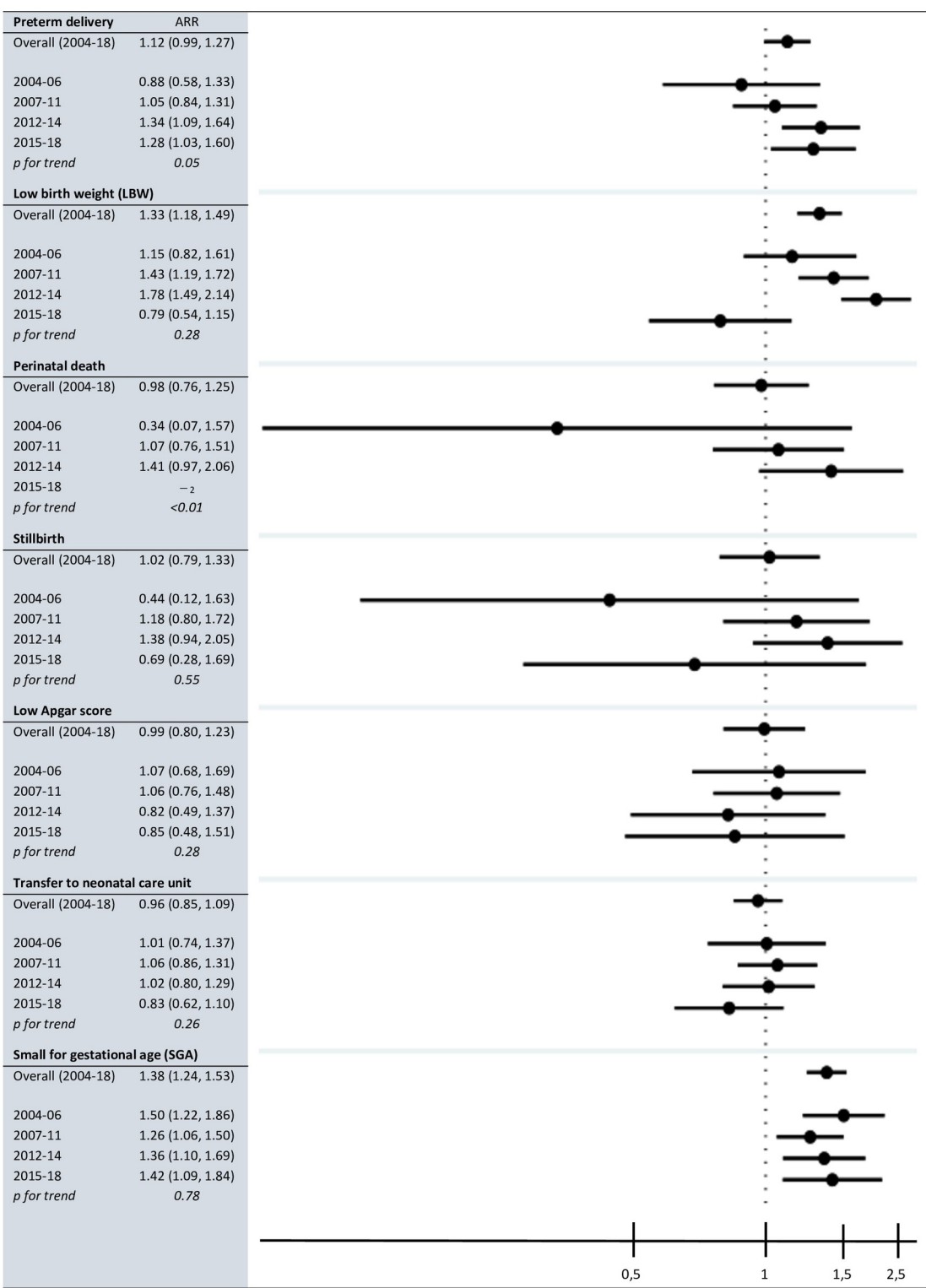

**Fig 3. Adverse pregnancy outcomes among HIV-positive and HIV-negative pregnant women in Northern Tanzania, overall adjusted relative risks and by time period₁, 2004–2018.** ARR: Adjusted relative risk. Forest plot shows overall adjusted relative risks in our data from 2004–2018, and in time period 2–5, HIV-positive women compared with HIV-negative women. X-axis has log scale, and ARRs are adjusted for maternal age, marital status, parity (not in the last time period), current residence and education level. P-values in table are for trends in ARRS. ₁ Time periods: 2000–2003: Pilot phase before national PMTCT guidelines (excluded from these analyses); 2004–2006: WHO 2004 guidelines; 2007–2011: Revised WHO 2004 guidelines; 2012–

2014: WHO Option A guidelines; 2015–2018: WHO Option B+ guidelines. ₂ Two variables that were used for defining early neonatal death were missing in the last time period.

birth weight the proportion among women with unknown HIV status still approximated the proportion among HIV-positive women (13.3% and 13.6%, respectively). Furthermore, the proportions of adverse perinatal outcomes increased among women with unknown HIV status from the second to the last time period. As HIV infection is a known risk factor for several of these adverse perinatal outcomes, we suspect that over time an increasing proportion of women with unknown HIV status were in fact HIV-positive. As we showed in a previous paper [30], there was a large decline in the proportion of women who were HIV-positive when estimated only among those with known HIV status, and we discussed that it might not represent a true decline as those selected for HIV tests at the beginning of the period most likely were at higher risk of infection. Since the proportion of women with unknown HIV status declined during all time periods, it is logical to assume that the true trend in HIV prevalence in our study sample was somewhere between being stable and declining. A decrease corresponds with national trends for young women (age 15 to 24) in Tanzania, which showed HIV prevalence of 4.1% in 2000 and 2.4% in 2018 [2]. The demographic data in the different time periods interestingly indicated a lower socioeconomic status for HIV-positive women and women with unknown HIV status compared with HIV-negative women. Low socioeconomic status, including single motherhood, low education and low income level are known barriers to seeking health care, and can thus be barriers to being tested for HIV during pregnancy as well [39]. It is plausible that women with unknown HIV status to an increasing extent might constitute a marginalized group of women.

## Trends in perinatal outcomes

We found that the overall risks of having a LBW baby and a SGA baby were higher for HIV-positive women compared with HIV-negative women, consistent with prior findings [5]. The observed trend for many of the studied outcomes might possibly be explained by changes in antiretroviral treatment (ART) coverage, duration of treatment, or type of ART regimen over time, although such associations remain speculations since we do not have detailed information about ART drugs and duration of ART in our data. Similar to many of the studied

**Table 3. Quantile regression analyses assessing annual change in standardized birth weight in grams (95% CI) for term deliveries from 2004 to 2018, stratified by maternal HIV status.** Birth weight variable is standardized for gestational age from completed 37 weeks to completed 42 weeks. Adjusted for maternal age, marital status, current residence and education level[1].

|  | HIV status | | All women |
|---|---|---|---|
|  | HIV- | HIV+ |  |
| **10th percentile** | 2.3 (-1.8, 6.4) | -11.4 (-35.3, 12.5) | 1.8 (-1.8, 5.4) |
| **25th percentile** | 4.3 (1.4, 7.3) | -5.4 (-19.7, 8.9) | 4.9 (2.2, 7.5) |
| **50th percentile** | 5.4 (2.7, 8.1) | -12.4 (-23.9, -1.0) | 6.0 (3.7, 8.4) |
| **75th percentile** | 6.0 (2.8, 9.2) | -8.8 (-22.4, 4.9) | 6.7 (3.7, 9.6) |
| **90th percentile** | 9.4 (5.4, 13.4) | 8.0 (-11.9, 27.9) | 10.2 (6.6, 13.8) |
| **Median birth weight in 2004 (grams)** | 3200 | 3150 | 3150 |

[1]Not adjusted for parity, as the parity variable was missing in most of the records in the last time period

outcomes, the ART coverage seems to have increased over time, but with a sudden decline from 2012–2014 to 2015–2018. The effect of ART on adverse perinatal outcomes remains controversial [15], but several previous studies indicate that women on combination antiretroviral treatment (cART) have a higher risk of having a preterm delivery and a LBW baby compared with women on monotherapy, while stillbirth and SGA are less frequently studied outcomes and thus the impact on the two outcomes is associated with greater uncertainty [11–14, 16–18]. Some studies report that protease inhibitor (PI)-based regimens, especially ritonavir-boosted PI therapy, in particular might be associated with having a preterm delivery when compared with monotherapy or non-boosted triple therapy [12, 13, 16, 17, 40]. Moreover, when cART initiated before pregnancy is compared with initiation during pregnancy, studies have found an increased risk of preterm delivery and LBW [41]. In the last time period zidovudine/azidothymidine (ZDV/AZT) was exchanged for tenofovir (TDF) in WHO Option B + first-line regimen. The observed decline in the ARRs for this time period corresponds with results from a systematic review where HIV-positive women receiving TDF- versus non-TDF-based ART during pregnancy were found to have a lower risk of stillbirth, and also a (tendency towards) slightly lower risk of having a preterm delivery, a LBW baby and a SGA baby [42].

Although the non-TDF-based ART in this systematic review included a variety of regimens, often unknown, several of these regimens were indeed ZDV/AZT-based. Drug-induced anemia is a common side effect of ZDV/AZT [43, 44], and although the etiology of anemia is multifactorial, anemia is a known factor contributing to fetal growth restriction [45, 46]. This might possibly have affected the proportions for several of our studied perinatal outcomes, as they could be affected by fetal growth [46].

## Information on ART drugs in the birth registry

We observed an increase in ART coverage over time, but did not have data on the specific drugs used as ART during pregnancy or timing of initiation. In supplementary analyses where we excluded HIV-positive women not on ART, the trend towards a higher relative risk in HIV-positive women compared with HIV-negative women from 2004–06 to 2012–14 appeared more evident for having a preterm delivery, a LBW baby, perinatal death and stillbirth. These findings support our suggestions that maternal use of ART drugs during these time periods might possibly have contributed to the increase in ARRs.

## Quantile regression, birth weight over time

We found a clear trend among babies born to HIV-negative women and all women towards increasing birth weight over time at all percentiles, and the highest increases per year in grams were at the upper percentiles. In contrast, an unfavourable trend was seen for babies born to HIV-positive women where there was an annual decline in the value of all percentiles except the 90th percentile. The decline in birth weight over time was only seen for babies born to the HIV-positive women among whom an increasing proportion were on ART and on a triple-drug regimen. Results from the analyses of birth weight percentiles correspond with the increasing proportion of LBW babies born to HIV-positive women over time, and with the increase in ARR of LBW for HIV-positive women compared with HIV-negative women, except for the last time period.

## Increase in preterm delivery

We observed an increase in the proportion having a preterm delivery for both HIV-negative and HIV-positive women, although the largest increase was seen for the HIV-positive women. First, an increase over time has also been seen in global and regional preterm percentages over

the last 20 years [47]. The estimates have been among the highest in sub-Saharan Africa, with a mean percentage of 12.3% in 2010 [48]. Both an increased proportion of caesarean sections and spontaneous preterm deliveries due to a variety of reasons, such as increases in maternal age and underlying maternal health problems like diabetes and high blood pressure, could explain the increased preterm delivery percentages [49]. Second, the increased proportion with preterm delivery among HIV-positive women could as mentioned be explained by increased ART coverage.

## Strengths and limitations

A major strength of our study is the large number of observations over 18 years that allowed us to stratify the analysis into several time periods and assess time trends. The data have been systematically collected by structured interviews by trained midwives on a daily basis since 2000, ensuring consistency in data quality over time. Lastly, we reduced any potential selection bias by excluding women living outside Moshi.

A limitation in our study is the substantial proportion of women with unknown HIV status in the two first time periods. Although we eliminated the first time period from several analyses, there was still 30.5% with unknown HIV status in the second time period. The missing observations could reduce the statistical power of our analyses. Our data did not include information about the types of drugs used as ART, CD4 cell count, clinical stage, timing of initiation of ART, duration of ART or consistency in ART usage. Although we have adjusted for potential risk factors, there may be other causes of adverse perinatal outcomes that are not available in the birth registry. As the majority of the HIV-positive women were on ART, there were too few untreated HIV-positive women to conduct comparative analyses for the treated versus untreated. In addition, we did not have data on HIV status for the babies, but it appears reasonable to assume that very few of the babies were HIV-positive considering the high proportion of women on treatment. It is therefore unlikely that the increased risk of adverse outcomes was mainly due to the babies being HIV-positive. We cannot exclude the possibility that selection mechanisms affect our results. Since KCMC is a tertiary referral facility, findings may not necessarily be generalizable to the general population, as the majority of severe cases are referred to deliver at KCMC. This accounts for both HIV-negative and HIV-positive women, and might lead to overestimation of adverse perinatal outcomes. Also, the general selection of women who deliver at Kilimanjaro Christian Medical Centre (KCMC) may be affected by possible cost-sharing fees at KCMC as it is a private hospital. There might as well be a selection of more HIV-positive women relative to HIV-negative women delivering at KCMC since we in a previous paper found that more HIV-positive than HIV-negative women were referred for delivery at this hospital [30].

## Conclusion

In our study population maternal HIV infection was associated with a higher risk of low birth weight and small for gestational age. Analyses of time trends in birth weight percentiles showed an annual increase in birth weight for babies born to HIV-negative women and a corresponding decline in birth weight for babies born to HIV-positive women. Increasing risks were seen from 2004–2006 to 2012–2014 for preterm delivery, low birth weight (LBW), perinatal death and stillbirth among HIV-positive women compared with HIV-negative women. From 2012–2014 to 2015–2018 there was a decline in the relative risks for these outcomes, the decline being most evident for having a LBW baby. Type of cART regimen might possibly explain some of these changes in risks, and future studies should explore possible associations.

## Supporting information

**S1 Table. Sociodemographic characteristics of study sample by HIV status and time period, 2000–2018.**
(DOCX)

**S2 Table. Adverse perinatal outcomes among pregnant women in Northern Tanzania, overall proportions and by HIV status and time period, 2004–2018.**
(DOCX)

**S3 Table. Relative risks from regression analyses (2004–2018), both main analyses included in forest plot and supplementary analyses.**
(DOCX)

## Acknowledgments

The authors thank the KCMC administration and all the staff at KCMC Reproductive Health Department for the work with interviews, registration of the data and maintenance of the Birth Registry through all these years. The authors also thank Truls Østbye for his contributions at the planning stage of the study.

## Author Contributions

**Conceptualization:** Tormod Rebnord, Blandina Theophil Mmbaga, Ingvild Fossgard Sandøy, Rolv Terje Lie, Anne Kjersti Daltveit.

**Data curation:** Blandina Theophil Mmbaga, Bariki Mchome, Michael Johnson Mahande.

**Formal analysis:** Tormod Rebnord, Ingvild Fossgard Sandøy, Rolv Terje Lie, Anne Kjersti Daltveit.

**Funding acquisition:** Anne Kjersti Daltveit.

**Investigation:** Tormod Rebnord, Anne Kjersti Daltveit.

**Methodology:** Tormod Rebnord, Ingvild Fossgard Sandøy, Rolv Terje Lie, Anne Kjersti Daltveit.

**Project administration:** Tormod Rebnord, Anne Kjersti Daltveit.

**Supervision:** Tormod Rebnord, Anne Kjersti Daltveit.

**Validation:** Tormod Rebnord.

**Visualization:** Tormod Rebnord, Anne Kjersti Daltveit.

**Writing – original draft:** Tormod Rebnord.

**Writing – review & editing:** Tormod Rebnord, Blandina Theophil Mmbaga, Ingvild Fossgard Sandøy, Rolv Terje Lie, Bariki Mchome, Michael Johnson Mahande, Anne Kjersti Daltveit.

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
