## [Decision Letter · Decision Letter 0]

8 Dec 2022

PONE-D-22-28889Time trends in perinatal outcomes among HIV-positive pregnant women in Northern Tanzania: A registry based studyPLOS ONE

Dear Dr. Rebnord,

Thank you for submitting your manuscript to PLOS ONE. After careful consideration, we feel that it has merit but does not fully meet PLOS ONE’s publication criteria as it currently stands. Therefore, we invite you to submit a revised version of the manuscript that addresses the points raised during the review process. 

The reviewers' comments are attached. Kindly review these comments, respond point-by-point and revise accordingly if agreeable. Please submit your revised manuscript by Jan 22 2023 11:59PM. If you will need more time than this to complete your revisions, please reply to this message or contact the journal office at plosone@plos.org. Please include the following items when submitting your revised manuscript:A rebuttal letter that responds to each point raised by the academic editor and reviewer(s). You should upload this letter as a separate file labeled 'Response to Reviewers'.A marked-up copy of your manuscript that highlights changes made to the original version. You should upload this as a separate file labeled 'Revised Manuscript with Track Changes'.An unmarked version of your revised paper without tracked changes. You should upload this as a separate file labeled 'Manuscript'.If applicable, we recommend that you deposit your laboratory protocols in protocols.io to enhance the reproducibility of your results. Protocols.io assigns your protocol its own identifier (DOI) so that it can be cited independently in the future. For instructions see: https://journals.plos.org/plosone/s/submission-guidelines#loc-laboratory-protocols. Additionally, PLOS ONE offers an option for publishing peer-reviewed Lab Protocol articles, which describe protocols hosted on protocols.io. Read more information on sharing protocols at https://plos.org/protocols?utm_medium=editorial-email&utm_source=authorletters&utm_campaign=protocols.

We look forward to receiving your revised manuscript.

Kind regards,

Chika Kingsley Onwuamah, Ph.D.

Academic Editor

PLOS ONE

Journal Requirements:

3. Please upload a copy of Figure 3, to which you refer in your text on page 18. If the figure is no longer to be included as part of the submission please remove all reference to it within the text.

Reviewers' comments:

Reviewer's Responses to Questions

**Comments to the Author**

1. Is the manuscript technically sound, and do the data support the conclusions?

Reviewer #1: No

Reviewer #2: Yes

Reviewer #3: Partly

2. Has the statistical analysis been performed appropriately and rigorously? 

Reviewer #1: No

Reviewer #2: Yes

Reviewer #3: Yes

3. Have the authors made all data underlying the findings in their manuscript fully available?

Reviewer #1: Yes

Reviewer #2: Yes

Reviewer #3: Yes

4. Is the manuscript presented in an intelligible fashion and written in standard English?

Reviewer #1: Yes

Reviewer #2: Yes

Reviewer #3: Yes

5. Review Comments to the Author

Reviewer #1: 1. Sample size - It is unclear why the final analysis sample consisted of 41,156 mother-baby pairs compared to 62,927 babies delivered. Criteria for exclusion would improve clarity.

2. There was no comment on repeat pregnancies within the sample and how these were treated.

3. Authors have clearly outlined limitations which make a strong case for questioning the conclusions. For example, from the second to the last period under consideration, there is no difference in outcomes between HIV positive and HIV negative women. Women with unknown HIV status experienced highest proportions of preterm, perinatal death, still birth, low Apgar score and transfer to neonatal care. Placed against the primary focus of study which is comparison of outcomes for HIV positives with negatives, more explanation is needed on the significance of women without known HIV status.

4. Parity was said to have been excluded in the analysis however, in other sections parity is discussed as part of analysis. This is unclear.

5. The linkage between population-level HAART coverage and outcome variables is weak.

6. Inclusion of factors such as ART regimen, duration on treatment etc.. would improve the quality of analysis.

Reviewer #2: Summary of the Research

The study assessed the trends in perinatal outcomes in singleton deliveries among HIV-positive and HIV-negative women in Moshi, Tanzania across 5 time periods from 2000 to 2018 using registry data, the time periods corresponding to different PMTCT guidelines in the country. Appropriate ethical approvals were obtained. The authors showed a robust analysis of the data with results showing an increasing trend of adverse perinatal outcomes among HIV-positive women.

There was an increasing trend in preterm delivery among the entire cohort across the years, with higher increases in HIV-positive women. While trends for low birthweight (LBW), small for gestational age (SGA), perinatal death, and stillbirth also increased among HIV+ women, they remained stable for LBW and stillbirth and decreased for SGA and perinatal death among HIV-negative women. There was also a general trend of increased birth weight at all percentiles among HIV-negative women across all percentiles, and decreased birthweight in all except the 90th percentile among HIV-infected women.

The authors conclude that maternal ART may play a role in the observed increase in adverse perinatal outcomes in HIV-infected women and recommend continued careful monitoring of adverse effects of antiretroviral therapy.

The strength of the study is the large number of well-documented observations over almost two decades enabling assessment from time periods corresponding to different recommendations from guidelines. A major limitation is the lack of inclusion clinical stage of the women with HIV, the different antiretroviral regimens as well as the time of commencement of ARVs with regard to index pregnancy. These would have helped to further explain the trend of increased adverse outcomes observed.

The study showed an increasing trend of adverse perinatal outcomes in women living with HIV in Tanzania, even in the era of HAART for all people living with HIV.

The study is well undertaken and written and recommended for publication after the issue highlighted below is addressed.

Discussion of Specific Areas for Clarification

1. In the methods section (outcome variables, lines 213 and 214) and the results section (trends in perinatal outcomes, table 2), perinatal death and stillbirth are documented as separate outcome variables. The authors document the absence of 2 parameters for establishing early neonatal death in the last time period and so left perinatal deaths blank for that time period, while stillbirths for the time period are documented I recommend that since perinatal deaths include stillbirths, it may be more appropriate to split the perinatal deaths into its two constituents of stillbirths and early neonatal deaths. The time period with missing data can then be left blank for early neonatal deaths.

Reviewer #3: The manuscript is well written and structured. Authors provided population based data from a region in Tanzania from mother infant pairs. I have general questions for the authors:

1. Is there a reason why only singletons were included in the analysis?

2. For the oral consent, was this documented or recorded? was the interview conducted in only English or any local language?

3. How did authors classify the women into HIV positive and negative? Is this based on a HIV test conducted during pregnancy or at labor? The timepoint of the test is important

4. Did authors collect other maternal data like years of HIV diagnosis for the HIV positive women and BMI or nutritional status of the women?

5. In general the concluding statement shouldn't be only that ARVs are linked to unfavorable outcomes in the infants as first of all the infants are HIV exposed which can alter their immunity and growth outcomes as reported in other studies.

6. PLOS authors have the option to publish the peer review history of their article (what does this mean?). If published, this will include your full peer review and any attached files.

Reviewer #1: No

Reviewer #2: No

Reviewer #3: **Yes: **Sophia Osawe

---

## [Author Response · Author response to Decision Letter 0]

30 Apr 2023

Please see separate document for our comments to reviewers.

---

## [Decision Letter · Decision Letter 1]

6 Jun 2023

PONE-D-22-28889R1Time trends in perinatal outcomes among HIV-positive pregnant women in Northern Tanzania: A registry based studyPLOS ONE

Dear Dr. Rebnord,

Thank you for submitting your manuscript to PLOS ONE. After careful consideration, we feel that it has merit but does not fully meet PLOS ONE’s publication criteria as it currently stands. Therefore, we invite you to submit a revised version of the manuscript that addresses the points raised during the review process.

Kindly consider the new comments from the reviewer. I am hopefully it will help improve the paper 

We look forward to receiving your revised manuscript.

Kind regards,

Chika Kingsley Onwuamah, Ph.D.

Academic Editor

PLOS ONE

Journal Requirements:

Reviewers' comments:

Reviewer's Responses to Questions

**Comments to the Author**

1. If the authors have adequately addressed your comments raised in a previous round of review and you feel that this manuscript is now acceptable for publication, you may indicate that here to bypass the “Comments to the Author” section, enter your conflict of interest statement in the “Confidential to Editor” section, and submit your "Accept" recommendation.

Reviewer #1: (No Response)

Reviewer #2: All comments have been addressed

Reviewer #3: All comments have been addressed

2. Is the manuscript technically sound, and do the data support the conclusions?

Reviewer #1: Yes

Reviewer #2: Yes

Reviewer #3: Yes

3. Has the statistical analysis been performed appropriately and rigorously? 

Reviewer #1: Yes

Reviewer #2: Yes

Reviewer #3: Yes

4. Have the authors made all data underlying the findings in their manuscript fully available?

Reviewer #1: Yes

Reviewer #2: Yes

Reviewer #3: Yes

5. Is the manuscript presented in an intelligible fashion and written in standard English?

Reviewer #1: Yes

Reviewer #2: Yes

Reviewer #3: Yes

6. Review Comments to the Author

Reviewer #1: 1. Lines 257 & 258: If unknown maternal HIV status was high in first period thus leading to its exclusion from all analyses, it is unclear why it is included in descriptive statistics for sociodemographic characteristics and outcome variables.

2. Lines 327-329: This section seems to suggest that unknown HIV status is a risk factor. Could this picture be due to having HIV positive mothers within this group? Other factors could also contribute to this picture.

3. Lines 425: Appears speculative if ART regimen were not explored.

4. What about the impact of other causes of adverse perinatal outcomes?

Reviewer #2: (No Response)

Reviewer #3: (No Response)

7. PLOS authors have the option to publish the peer review history of their article (what does this mean?). If published, this will include your full peer review and any attached files.

Reviewer #1: No

Reviewer #2: No

Reviewer #3: **Yes: **Sophia Osawe

---

## [Author Response · Author response to Decision Letter 1]

17 Jul 2023

See attached file "Response to reviewers"

---

## [Decision Letter · Decision Letter 2]

26 Jul 2023

Time trends in perinatal outcomes among HIV-positive pregnant women in Northern Tanzania: A registry based study

PONE-D-22-28889R2

Dear Dr. Rebnord,

We’re pleased to inform you that your manuscript has been judged scientifically suitable for publication and will be formally accepted for publication once it meets all outstanding technical requirements.

Kind regards,

Chika Kingsley Onwuamah, Ph.D.

Academic Editor

PLOS ONE

Additional Editor Comments (optional):

Reviewers' comments:

Reviewer's Responses to Questions

**Comments to the Author**

1. If the authors have adequately addressed your comments raised in a previous round of review and you feel that this manuscript is now acceptable for publication, you may indicate that here to bypass the “Comments to the Author” section, enter your conflict of interest statement in the “Confidential to Editor” section, and submit your "Accept" recommendation.

Reviewer #1: All comments have been addressed

2. Is the manuscript technically sound, and do the data support the conclusions?

Reviewer #1: (No Response)

3. Has the statistical analysis been performed appropriately and rigorously? 

Reviewer #1: (No Response)

4. Have the authors made all data underlying the findings in their manuscript fully available?

Reviewer #1: (No Response)

5. Is the manuscript presented in an intelligible fashion and written in standard English?

Reviewer #1: (No Response)

6. Review Comments to the Author

Reviewer #1: (No Response)

7. PLOS authors have the option to publish the peer review history of their article (what does this mean?). If published, this will include your full peer review and any attached files.

Reviewer #1: No

---

## [Editor Report · Acceptance letter]

31 Jul 2023

PONE-D-22-28889R2 

Time trends in perinatal outcomes among HIV-positive pregnant women in Northern Tanzania: A registry-based study 

Dear Dr. Rebnord:

I'm pleased to inform you that your manuscript has been deemed suitable for publication in PLOS ONE. Congratulations! Your manuscript is now with our production department. 

Kind regards, 

on behalf of

Dr. Chika Kingsley Onwuamah 

Academic Editor

PLOS ONE